# SGLT2 Inhibitors in the Treatment of Diabetic Kidney Disease: More than Just Glucose Regulation

**DOI:** 10.3390/pharmaceutics15071995

**Published:** 2023-07-20

**Authors:** Jasna Klen, Vita Dolžan

**Affiliations:** 1Division of Surgery, Department of Abdominal Surgery, University Medical Centre Ljubljana, 1000 Ljubljana, Slovenia; jasna.klen@gmail.com; 2Department of Internal Medicine, Faculty of Medicine, University of Ljubljana, 1000 Ljubljana, Slovenia; 3Pharmacogenetics Laboratory, Institute of Biochemistry and Molecular Genetics, Faculty of Medicine, University of Ljubljana, 1000 Ljubljana, Slovenia

**Keywords:** diabetes mellitus, type 2, diabetic nephropathies, sodium–glucose contransporter-2 inhibitors, biomarkers

## Abstract

Diabetic kidney disease (DKD) is a severe and common complication and affects a quarter of patients with type 2 diabetes mellitus (T2DM). Oxidative stress and inflammation related to hyperglycemia are interlinked and contribute to the occurrence of DKD. It was shown that sodium–glucose cotransporter-2 (SGLT2) inhibitors, a novel yet already widely used therapy, may prevent the development of DKD and alter its natural progression. SGLT2 inhibitors induce systemic and glomerular hemodynamic changes, provide metabolic advantages, and reduce inflammatory and oxidative stress pathways. In T2DM patients, regardless of cardiovascular diseases, SGLT2 inhibitors may reduce albuminuria, progression of DKD, and doubling of serum creatinine levels, thus lowering the need for kidney replacement therapy by over 40%. The molecular mechanisms behind these beneficial effects of SGLT2 inhibitors extend beyond their glucose-lowering effects. The emerging studies are trying to explain these mechanisms at the genetic, epigenetic, transcriptomic, and proteomic levels.

## 1. Introduction

The prevalence of diabetes mellitus has reached epidemic proportions worldwide [1]. At the moment, 537 million adults from 20 to 79 years are living with diabetes of any type. The projection shows us that we can expect 783 million patients with diabetes in 2045. It is known that 541 million adults have impaired glucose tolerance, which could lead to type 2 diabetes mellitus (T2DM). A substantial portion of patients with diabetes mellitus remain undiagnosed. Moreover, late complications of diabetes mellitus could be the first sign of the disease. Hyperglycemia is causally related to microvascular complications, while dyslipidemia, hypertension, smoking, and genetics also play an important role. Patients with T2DM have a high prevalence of microvascular complications, such as diabetic retinopathy, neuropathy, and diabetic kidney disease (DKD). A diagnosis of chronic kidney disease (CKD) is characterized by persistence of elevated urinary albumin excretion (albuminuria), low estimated glomerular filtration rate (eGFR), or associated with other symptoms of kidney damage, such as hematuria and morphological pathological abnormalities persisting for a duration of three months or longer [2,3]. DKD is a severe and frequent complication of diabetes mellitus and has an incidence of 25% in T2DM patients [4]. It is the leading cause of end-stage kidney disease (ESKD), which is rising steeply and leads to significant morbidity and cardiovascular mortality [5]. According to recent estimates, the number of patients undergoing kidney replacement therapy (KRT) worldwide reached 3.9 million, with high-income countries having the highest prevalence [6]. The main factor contributing to the onset of DKD is hyperglycemia; therefore, aiming to maintain HbA1c < 6.5% should be the major goal of primary interventions [7]. Importantly, it has been reported that stringent glycemic control decreased the incidence of albuminuria by 50%, and this favorable effect was maintained for more than ten years after the trial’s conclusion [8]. Similar results were observed in patients with T2DM, with each 1% decline in HbA1c decreasing the risk of microvascular complications by 37%, especially the likelihood of microalbuminuria [9]. Although normalization of glycemia may slow progression of DKD, it cannot halt it entirely [1]. One of the important factors for the onset and progression of CKD is also hypertension [10]. Normotensive patients with advanced CKD have a slower progression of kidney disease than hypertensive patients [1]. Due to conflicting study results, it is still unclear which blood pressure values could be recommended for patients with diabetes mellitus, with the generally recommended levels of blood pressure being below 140/90 mmHg [10]. According to the ESC/EASD guidelines, patients with diabetes who have several risk factors or even just one type of organ damage are in the same very high cardiovascular risk group as patients taking secondary prevention with developed cardiovascular disease. The Nephropathy In Diabetes type 2 (NID-2) study, which was multi-center, cluster-randomized, open-label, and focused on primary cardiovascular prevention, demonstrated in T2DM patients with albuminuria and diabetic retinopathy that multifactorial intensive treatment may have lower major adverse cardiovascular events and overall mortality. Based on a post hoc analysis, they found that a higher number of risk factors achieving the target level is associated with better cardiovascular-free survival in T2DM patients [11].

Angiotensin-converting enzyme (ACE) inhibitors or angiotensin receptor blockers (ARBs), which are RAAS blockers, demonstrated benefits in preventing CKD progression and are, therefore, considered as first-line antihypertensive drugs in patients with diabetes mellitus, hypertension, eGFR 60 mL/min/1.73 m^2^, and urine albumin-to-creatinine ratio (UACR) 300 mg/g Cr. Their good renoprotective effect may be due to the decrease in elevated intraglomerular pressure. They moderately lower the risk of albuminuria and reduce progression from CKD to ESRD [1,10,12]. Even with rigorous blood pressure control with ACE inhibitors or ARBs and strict glycemia control, only marginal success was achieved in lowering the risk for CKD, but the progression of CKD to ESKD and related mortality could be not halted [1]. On the other hand, it was found that adding non-steroidal mineralocorticoid receptor antagonists to the aforementioned therapy reduces proteinuria. Nevertheless, we must be very careful when prescribing non-steroidal mineralocorticoid receptors antagonists because of side effects, especially hyperkalemia. It has been shown that aldosterone has the ability to increase fibrosis and inflammation in addition to its role in controlling sodium balance via activating mineralocorticoid receptors [1].

The conventional treatments to maintain blood glucose levels do not always prevent DKD [13]. On the other hand, it was reported that sodium–glucose cotransporter-2 (SGLT2) inhibitors, which are quite novel but already a widely used therapy, may prevent the development and alter the natural progression of DKD by inducing systemic and glomerular hemodynamic changes, providing metabolic advantages, and diminishing inflammatory and oxidative stress pathways [1,14,15,16,17], which we have shown in the Figure 1.

The aim of this narrative review was to summarize the current knowledge on the role of SGLT2 inhibitors in the treatment of DKD and various physiological, clinical, and genetic factors that may influence the treatment outcomes.

## 2. Pathophysiology of CKD in Patients with T2DM

The pathophysiology of DKD is mostly driven by elevated reactive oxygen species (ROS) caused by prolonged hyperglycemia [18]. The main sources of ROS, such as mitochondrial respiratory chain dysfunction, lipoxygenase, xanthine oxidase, uncoupled nitric oxide synthase, NADPH oxidase (Nox), and polyol pathway and advanced glycation end products (AGEs), are indirectly or directly linked to hyperglycemia [19,20]. The enzyme Nox 4, which is expressed in different organs, plays the main role in the production of ROS in kidneys [21]. Glomerulus is sensitive to oxidative injury and oxidative stress may cause direct damage to podocytes, mesangial, epithelial, tubular, and endothelial cells, which may result in proteinuria and tubulointerstitial fibrosis [20,22]. The negative synergistic impact resulting from oxidative stress manifests in metabolic and hemodynamic alterations within the kidney. Oxidative stress may lead to increased release of angiotensin II, which is not only a hemodynamic actor but may also act as a cytokine. The kidneys have a local renin–angiotensin–aldosterone system (RAAS) and are independent of systemic RAAS; therefore, they can maintain increased levels of intrarenal angiotensin II. Furthermore, hyperglycemia in mesangial cells may stimulate the synthesis of angiotensin II and also renin [23]. Angiotensin II has been shown to elevate glomerular capillary pressure and permeability, induce kidney cell proliferation and hypertrophy, trigger the synthesis of cytokines and extracellular matrix (ECM), and facilitate the infiltration of macrophages and inflammation [24].

IL-18, IL-6, and tumor necrosis factor (TNF-*α*) may also be involved in the development and progression of DKD [25]. IL-6 levels were significantly increased in T2DM patients [26] and correlated with both the severity of albuminuria and morphological changes [27,28]. Chemoattractant cytokines and adhesion molecules, which act as the primary mediators of kidney injury, have the ability to increase the migration of circulating leukocytes towards the kidney and assist them in infiltrating the kidney tissues. Moreover, these cells also contribute to the development and progression of kidney injury by releasing cytokines and other mediators that strengthen and perpetuate the inflammatory response [29]. Nuclear factor-*κ*B (NF-*κ*B) is another factor that is important in inflammation. This is a ubiquitous transcription factor that could be triggered by hyperglycemia or AGEs. An increasing body of evidence suggests that JAK-STAT (Janus kinase (JAK)-signal transducer and activator of transcription (STAT)) regulates the major pathway for processing inflammatory signals and plays a key role in the pathogenesis of DKD. As the disease progresses, tubulointerstitial expression of a variety of JAK and STAT isoforms increases, and this expression is inversely related to eGFR. The JAK-STAT pathway triggers expression of intracellular cytokines, which function as a bridge between the paracrine stimulation system and nuclear receptors. In addition, cytokines and hyperglycemia can activate key mechanisms that regulate cell activation, proliferation, recruitment, migration, and differentiation [30]. Furthermore, protein kinase C (PKC)-*α* and PKC-*β* isoforms activation was associated with increased NADPH activity and production of NADPH-dependent superoxide [31]. Increased angiotensin II levels, PKC activation, and expression of transforming growth factor beta (TGF-*β*) lead to synthesis of the mesangial matrix. The increased levels of angiotensin II also activate NADPH oxidase, which increases ROS production in the kidney. On the other hand, TGF-*β* stimulates cellular differentiation and proliferation, hormone secretion, immune function, and is responsible for excessive remodeling of the extracellular matrix in the mesangium and for increased fibrotic processes in the tubular interstitial space [23]. Several studies have shown that oxidative stress and inflammation are linked and work hand in hand in the occurrence of CKD, but it is difficult to point out the primary irregularity [32,33,34,35]. The increased TNF-*α* expression in the epithelial cells of the proximal tubule and glomerulus enhances the expression of cytokines that act on cell survival, proliferation, adhesions, inflammatory responses, and apoptosis [34,35]. Furthermore, TNF-*α* may decrease GFR and causes vasoconstriction due to increased endothelin-1 (ET-1) production and also impairs the glomerular filtration barrier and proteinuria.

## 3. SGLT2 Transporters and SGLT2 Inhibitors in the Treatment of Diabetic Kidney Disease

Sodium-driven glucose symporters (SGLTs) are solute carriers facilitating glucose transport. Among the six SGLTs transporters, SGLT2 is responsible for 90% of glucose reabsorption and also for the majority of sodium reabsorption in kidneys. It has a low affinity and high capacity for glucose and is expressed in the brush border membrane of the S1 segment in the kidney cortex [36]. Many pathophysiological abnormalities are associated with increased SGLT2 activity in the proximal tubule. SGLT2 inhibitors reverse many of these abnormalities and significantly slow down DKD progression [1]. The administration of SGLT2 inhibitors is associated with proximal tubular natriuresis, which helps to restore distal sodium delivery and tubulo-glomerular feedback. This, in turn, leads to afferent arteriolar vasoconstriction, thereby reducing renal hyperfiltration and glomerular pressure, which lowers albuminuria levels. In T2DM patients, long-term treatment with SGLT2 inhibitors as opposed to placebo resulted in a slower decrease in GFR and a 30–50% reduction in albuminuria. The levels of albuminuria are acknowledged as a surrogate indicator of the advancement of kidney disease. The randomized double-blind RED trial found that SGLT2 inhibitors lower the measured GFR and filtration fraction and do not increase kidney vascular resistance [1,14,15,16,17]. The effects of SGLT2 inhibition on systemic hemodynamics may further contribute to kidney protection through changes in glomerular hemodynamics. This inhibition reduces plasma volume and lowers systolic blood pressure by approximately 3–6 mm Hg and diastolic blood pressure by approximately 1–2 mm Hg regardless of the individual’s hypertension status, leading to the activation of the sympathetic nervous system and RAAS and reduction in arterial stiffness. These effects may also be observed in patients with low eGFR [37,38,39]. The role of the SGLT1 and SGLT2 transporters within the nephron and SGLT2 inhibitors’ action are graphically presented in Figure 2.

Despite the fact that glucosuria reduces the mean plasma glucose concentrations and ameliorates glucotoxicity, resulting in improved cell function and insulin sensitivity [1], it is known that SGLT2 inhibitors have a mild impact on glycemic control in patients with T2DM who have normal kidney function. This outcome is achieved by stimulating the excretion of glucose with inhibition of the SGLT2 transporter in the S1 segment. Moreover, the accompanying reduction in caloric balance leads to an average weight loss of around 1–3 kg [40]. Although some of the weight loss can be attributed to increased sodium excretion, the primary cause is a decrease in body fat mass, and this effect is more pronounced in T2DM patients who have higher baseline hemoglobin A1c levels [41,42]. Additionally, glucosuria results in a metabolic shift towards a fasting state, which is characterized by an increase in the utilization of lipids and ketones as energy substrates. The metabolic shift caused by SGLT2 inhibitors may enhance energy utilization efficiency and stimulate low-energy cellular sensors, resulting in decreased hypoxia and improved mitochondrial function at both the cellular and organ levels [43,44]. It was found that SGLT2 inhibitors provide kidney protection by reducing inflammation and oxidative stress, as shown by changes in biomarkers related to cytokine/chemokine profiles and AGEs [45]. Recent studies of SGLT2 inhibitors have demonstrated improvements in endothelial function and reductions in local ROS generation [46,47]. Lee et al. found, in the cultured proximal tubular cells, that empagliflozin enhanced mitochondrial biogenesis and the balance of mitochondrial fission and fusion proteins, reducing ROS and also decreasing expression of TGF-β [48]. Additionally, they consistently reduce tubular injury markers and inflammatory mediators, including interleukin-6, nuclear factor-κB, kidney injury molecule 1, and profibrotic factors, such as TGF-β and fibronectin [13,49,50,51]. SGLT2 inhibitors may also attenuate kidney hypoxia by reducing the energy expenditure involved in tubular sodium and glucose reabsorption, leading to increases in erythropoietin production and hematopoiesis [49,52,53].
Figure 2The location of SGLT1 and SGLT2 transporters within the nephron and SGLT2 inhibitors’s action in glucose excretion by the kidneys.
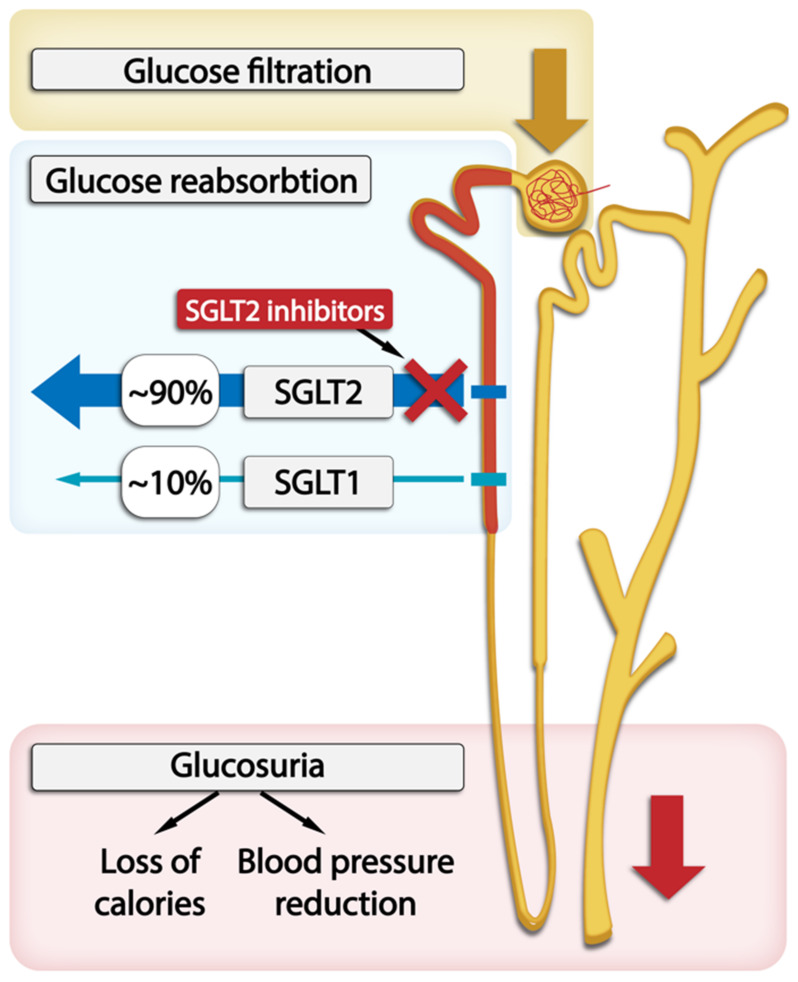



It needs to be pointed out that T2DM alone may also be an independent risk factor for acute kidney injury (AKI). AKI is more frequent in T2DM patients when they are undergoing surgery, taking some specific medications, or have sepsis or septic shock. On the other hand, AKI may also manifest as acute exacerbation of DKD. There has been a continuous concern about potential negative effects of SGLT2 inhibitors on the risk and prognosis of AKI due to the changes in kidney function associated with their usage, such as significant hypovolemia and different types of diabetic ketoacidosis (DKA). However, the results from clinical trials, observational studies, and meta-analyses have consistently shown no increase in the risk of AKI or negative effects of AKI associated with SGLT2 inhibitors. On the contrary, SGLT2 inhibitors were reported to reduce the risk of AKI by 30–40% [54]. In terms of CKD, SGLT2 inhibitors’ capacity to halt the loss of GFR with aging may be associated with a lower risk and better prognosis of AKI. However, for individuals with an already significantly low eGFR, the initial decrease in eGFR commonly observed when starting treatment with SGLT2 inhibitors may further increase the likelihood of experiencing an acute adverse event. Therefore, the recommendations clearly state that we can introduce SGLT2 inhibitors when eGFR is ≥45 mL/min, and they are not endorsed or recommended in ESRD. Accordingly, in patients who undergo AKI, it is advisable to stop the use of an SGLT2 inhibitor to prevent possible worsening of low plasma volume, low blood pressure, or low glomerular perfusion [55].

## 4. Treatment with SGLT2 Inhibitors and Kidney Outcomes

Several large clinical trials investigated clinical outcomes of treatment with SGLT2 inhibitors; however, kidney outcomes were mostly evaluated as secondary endpoints (Table 1). On the other hand, some of these studies also included patients without T2DM.

The CREDENCE (Canagliflozin and Renal Events in Diabetes with Established Nephropathy Clinical Evaluation) study was the first double-blind, randomized, placebo-controlled clinical trial that included 4401 patients with T2DM and albuminuric CKD treated with ACE inhibitors or ARBs. All subjects had HbA1c levels from 6.5 to 12.0%, eGFR of 30 to <90 mL/min per 1.73 m^2^, albuminuria (30–500 mg/mmol), and were randomized into a canagliflozin or placebo group. The composite outcome included either ESKD (dialysis, transplantation, or an estimated GFR of 15 mL/min per 1.73 m^2^ sustained for at least 30 days based on central laboratory assessment) or a doubling of serum creatinine level from baseline (average of randomization and prerandomization) sustained for at least 30 days, or a death caused by kidney or cardiovascular disease. The study was stopped early because of clearly demonstrated superiority of canagliflozin, showing a 30% relative reduction in the primary kidney outcome by canagliflozin (HR: 0.70; 95% CI: 0.59–0.82). In addition, there was significant evidence of benefits in regard to the secondary outcome of kidney transplant, dialysis, or kidney-related mortality (HR: 0.72; 95% CI: 0.54–0.97) [56]. Two double-blind, randomized studies were conducted concurrently as a part of the CANagliflozin cardioVascular assessment study (CANVAS) program. The CANVAS and CANVAS-R (renal) studies included 10,142 patients with or without baseline CKD. A 27% lower risk in progression of albuminuria (HR:0.73; 95% CI: 0.67–0.79) and a 40% (HR: 0.60; 95% CI: 0.47–0.77) lower risk for a composite kidney outcome consisting of ≥40% reduction in eGFR, need for KRT, or death from kidney cause were observed in the canagliflozin group compared to placebo [57].

The DECLARE-TIMI 58 (Dapagliflozin Effect on Cardiovascular Events-Thrombolysis in Myocardial Infarction 58) study was predominantly designed to evaluate the cardiovascular safety of dapagliflozin. The composite outcome consisting of a ≥40% decrease in eGFR to <60 mL/min per 1.73 m^2^, kidney failure, and death from cardiovascular or kidney cause was 4.3% in patients on dapagliflozin and 5.6% in the placebo group (HR: 0.76; 95% CI: 0.67–0.87) [58]. The cardiovascular safety of dapagliflozin was also the primary outcome in the DAPA-HR (Dapagliflozin and Prevention of Adverse-outcomes in Heart Failure) study. After a period of 18.2 months, a sustained ≥50% reduction in eGFR, kidney failure, or death from kidney cause in dapagliflozin and in the placebo arm were 1.2% and 1.6%, respectively (*p* = 0.17) [59]. The DELIGHT (An Exploratory Phase II/III, Randomized, Double-blind, Placebo Controlled, Parallel Design Study to Evaluate the Efficacy, Safety and Pharmacodynamics of Dapagliflozin and Dapagliflozin in Combination With Saxagliptin in CKD Patients With Type 2 Diabetes Mellitus and Albuminuria Treated With ACEi or ARB) study enrolled T2DM patients with HbA1c of 7–11%, UACR 30–3500 mg/g, and eGFR of 25–75 mL/min per 1.73 m^2^ who were on a stable dose of ACE inhibitors or ARBs. The patients were randomized into three treatment arms: 145 were treated with dapagliflozin, 155 with dapagliflozin and saxagliptin, and 148 received a placebo and were followed up every 4 weeks. Within the entire study period, reduced UACR was observed in the dapagliflozin and also in the dapagliflozin–saxagliptin arm. At the end of the study, which lasted 24 weeks, the difference in the mean UACR change from baseline was −21% (*p* = 0.011) in the dapagliflozin and −38% (*p* < 0.0001) in the dapagliflozin–saxagliptin arm. Furthermore, eGFR reductions were also observed at the end of the study, with differences in the mean change from baseline of –2.4 mL/min per 1·73 m^2^ (*p* = 0.011) in the dapagliflozin arm and –2.4 mL/min per 1.73 m^2^ (*p* = 0.0075) in the dapagliflozin–saxagliptin arm when compared to placebo [60]. A randomized, double-blind, placebo-controlled, multi-center clinical study DAPA-CKD (The Dapagliflozin and Prevention of Adverse outcomes in Chronic Kidney Disease) enrolled 4094 patients with/without T2DM on a stable dose of an ACE inhibitor or ARBs. The eligible subjects had eGFR 25–75 mL/min per 1.73 m and UACR of 200–5000 mg/g. The primary composite outcome, which included a sustained decline in eGFR of at least 50%, kidney failure, or death from kidney or cardiovascular causes, was reduced by 39% with dapagliflozin (HR: 0.61; 95% CI: 0.51–0.72). These results were comparable in patients with and without T2DM [61].

EMPA-KIDNEY (Study of Heart and Kidney Protection With Empagliflozin) was also a double-blind, randomized, placebo-controlled study with a median duration of 2 years, which included 6600 patients with CKD and eGFR of ≥20 to <45 mL/min per 1.73 m^2^, or an eGFR of ≥45 to <90 mL/min per 1.73 m^2^ with UACR of at least 200 mg/g. Patients with or without T2DM treated with ACE inhibitors or ARBs were randomized into two groups. The patients in the first group were receiving 10 mg of empagliflozin, while the patients in the other group were receiving a placebo. The primary outcome was a composite of death from kidney or cardiovascular causes, ESKD, a persistent decline in eGFR <10 mL/min per 1.73 m^2^, and a sustained decline in eGFR of less than 40% from the baseline. The progression of kidney disease or death from cardiovascular causes appeared in 13.1% of the patients on empagliflozin and in 16.9% of the patients in the placebo group (HR, 0.72; 95% CI, 0.64 to 0.82; *p* < 0.001). There were no significant differences between both groups regarding the composite outcome of hospitalization for heart failure, death from cardiovascular causes, or death from any cause. However, the rate of hospitalization from any cause was lower in the empagliflozin group compared to the placebo group (HR, 0.86; 95% CI, 0.78 to 0.95; *p* = 0.003). Both groups experienced equal rates of major adverse events [62]. The EMPA-REG (Empagliflozin Cardiovascular Outcome Event Trial in Type 2 Diabetes Mellitus Patients-Removing Excess Glucose) study included patients with advanced kidney disease with eGFR of 30 mL/min or more. Empagliflozin slowed the progression of kidney failure, and there were fewer clinically relevant kidney events in this group [63].

The EMPEROR-Reduced (Empagliflozin Outcome Trial in Patients with Chronic Heart Failure with Reduced Ejection Fraction) study included 3730 patients with/without T2DM and heart failure with an ejection fraction of 40% or less who had appropriate treatment for heart failure. The median follow-up at 21 months showed that annual reduction in eGFR was slower in the empagliflozin group than in the placebo group (*p* < 0.001). Furthermore, patients treated with empagliflozin had a lower risk of serious kidney outcome [64]. The EMPEROR-Preserved (Empagliflozin Outcome Trial in Patients with Chronic Heart Failure with Preserved Ejection Fraction) study was a double-blind and randomized clinical trial that included 5988 patients with class II–IV heart failure and an ejection fraction of more than 40% who were randomized in receiving empagliflozin 10 mg or a placebo in addition to their standard therapy. The median follow-up at 26.2 months indicated a slower rate of decline in the eGFR in the empagliflozin group (*p* < 0.001) [65]. In the EMPEROR-Reduced and the EMPEROR-Preserved study, the HR values for major kidney events were 0.51 (95% CI, 0.33 to 0.79) and 0.95 (95% CI, 0.73 to 1.24), respectively [66].

The VERTIS-CV (Evaluation of Ertugliflozin Efficacy and Safety Cardiovascular Outcomes) study included T2DM patients treated with ertugliflozin or a placebo. Although UACR was reduced, the study did not observe significant changes in composite kidney outcomes. eGFR was reduced after ertugliflozin treatment, but the values returned to baseline and were higher after 104 weeks [67].

The SCORED (Effect of Sotagliflozin on Cardiovascular and Renal Events in Patients with Type 2 Diabetes and Moderate Renal Impairment Who Are at Cardiovascular Risk) study included a total of 10.584 T2DM patients with CKD (eGFR, 25 to 60 mL per minute per 1.73 m^2^) that were randomized into two study arms: 5292 were treated with sotagliflozin and 5292 received a placebo. Although this study was primarily designed to assess cardiovascular safety, the secondary end points included a composite kidney outcome of the first occurrence of a sustained decrease of ≥50% in the eGFR from baseline for ≥30 days, long-term dialysis, kidney transplantation, or sustained eGFR of <15 mL/min/1.73 m^2^ for ≥30 days. The kidney outcomes did not differ significantly between the two groups [68].

To summarize the kidney outcomes reported by all these randomized clinical studies, we can conclude that, in T2DM patients with or without prevalent cardiovascular diseases, SGLT2 inhibitors may reduce albuminuria, progression of DKD, doubling of serum creatinine levels, and initiation of KRT to more than 40%.

The randomized data support the use of SGLT2 inhibitors in patients with CKD or heart failure regardless of diabetes status, primary kidney disease, or kidney function in order to reduce the risk of kidney disease progression and acute kidney injury [69,70].

Furthermore, we expect very promising results from the treatment of DKD with the novel and more selective non-steroidal mineralocorticoid receptor antagonist finerenone, which has more potent anti-inflammatory and anti-fibrotic effects on the kidney than spironolactone [71].
pharmaceutics-15-01995-t001_Table 1Table 1Comparison of kidney outcome studies.StudyStudy DesignNo. of PatientsDrug and Dosage (mg)/ComparatorMedian Follow-Up (Months)eGFR (mL/min/1.73 m^2^)UACR (mg/g)OutcomesRef.CREDENCERCT4.401canagliflozin 100/placebo31.230–59>300HR 0.66 (95% CI 0.53–0.81); *p* < 0.001 for canagliflozin group, indicating a 34% lower relative risk for the composite outcome of end-stage kidney disease; doubling of serum creatinine level or death from kidney causes[53]CANVASRCT10.142canagliflozin 100/300/placebo2930–59>300HR 0.60 (95% CI 0.47–0.77) for canagliflozin group, indicating a 40% lower relative risk for the composite outcome of end-stage kidney disease; doubling of serum creatinine level or death from kidney causes[54]DAPA-CKDRCT4.000dapagliflozin 10/placebo28.825–45>1000HR 0.56 (95% CI 0.45–0.68) fort he composite composite of decline in eGFR of ≥50%, ESKD, or death from kidney causes[58]DAPA-HFRCT4.744dapagliflozin 10/placebo18.230–59/HR 0.71 (95% CI 0.44–1.16) for the composite of worsening kidney function, i.e., reduction of ≥50% in the eGFR sustained for ≥28 days; ESKD, or death from kidney causes[56]DECLARE TIMI-58RCT17.160dapagliflozin 10/placebo50.4<60>300HR 0.53 (95% CI 0.43–0.66), *p* < 0.0001 for dapagliflozin group, indicating 47% lower relative risk for the composite outcome of end-stage kidney disease; doubling of serum creatinine level or death from kidney causes[55]DELIGHTRCT1.187dapagliflozin 10/dapagliflozin 10 + saxagliptin 2.5/placebo625–7530–3500?The difference in mean UACR change from baseline was −21.0% (95% CI −34.1 to −5.2; *p* = 0·011) for dapagliflozin and −38.0% (−48.2 to −25.8; *p* < 0.0001 for dapagliflozin–saxagliptin[57]EMPA-REGRCT7.020empagliflozin 10/25/placebo37.230–59>300HR 0.61 (95% CI 0.53–0.70), *p* < 0.001 in empagliflozin group, indicating 39% lower relative risk for the composite outcome of end-stage kidney disease; doubling of serum creatinine level or death from kidney causes[60]EMPEROR reduced and preservedRCT9.718empagliflozin 10/placebo1620–59>300HR 0.50 (95% CI 0.32–0.77) and HR 0.95 (95% CI 0.73–1.24) for the composite kidney outcome of chronic dialysis or kidney transplantation or sustained reduction by ≥40% in eGFR or sustained eGFR[61,62]EMPA-KIDNEYRCT6.600empagliflozin 10/placebo2420–45 or 45–90≥200HR 0.72 (95% CI 0.64 to 0.82; *p* < 0.001) for end-stage kidney disease (ESKD), a sustained decline in eGFR to less than 10 mL per minute per 1.73 m^2^; a sustained decline in eGFR of at least 40% from baseline, or death from kidney causes[59]VERTIS-CVRCT8.246ertugliflozin 15/5/placebo4230–59>300HR 0.81 (95% CI 0.63–1.04) in composite kidney outcome: doubling of serum creatinine, kidney replacement therapy, or death from kidney causes[64]SCOREDRCT10.584sotagliflozin 200 or 400/placebo16/15.925–59>300HR 0.71 (95% CI 0.46–1.08) for the composite outcome of first occurrence of sustained decrease of ≥50% in eGFR from baseline for ≥30 days, long-term dialysis, kidney transplantation, or sustained eGFR 15 mL/min/1.73 m^2^ for ≥ 30 days[65]


## 5. Molecular Mechanisms and Molecular Genetics of SGLT Action—Potential Biomarkers of Treatment

### 5.1. Genetic and Genomic Factors Related to SGLT2 Action

SGLT2 expression levels and activity may be directly related to genetic variability in its coding gene *SLC5A2* (solute carrier family 5 member 2). Rare loss-of-function *SLC5A2* mutations were associated with familial renal glucosuria, a benign condition characterized by decreased renal glucose reabsorption and increased glucose excretion in the presence of normal blood glucose levels [72]. Several common genetic polymorphisms were also reported in *SLC5A2,* with the initial studies suggesting their influence on glucose homeostasis in non-diabetic subjects [73]. We have reported that, in T2DM patients, *SLC5A2* rs9934336 was significantly associated with increased fasting blood glucose levels and HbA1c levels. This polymorphism was also significantly associated with higher risk for diabetic retinopathy, but not with DKD risk [74]. Another study observed no significant associations of *SLC5A2* polymorphisms with the metabolic traits of T2DM patients nor with their response to treatment with empagliflozin [75].

When considering the broader genetic background that may influence the response to SGLT2 inhibitors, one may first focus on the interaction of the treatment with the molecular mechanisms of the disease itself and, next, its complications. Genome-wide association studies (GWAS) enabled identification of more than 300 genetic susceptibility loci associated with T2DM (recently reviewed in [76]). However, T2DM complications may have a broader genetic background as they may be due to both glucose-dependent and glucose-independent mechanisms. This is reflected in the early family studies that confirmed the genetic heritability of CKD related to T2DM complications but have shown a lack of correlation with the severity of nephropathy [77]. This phenotypic diversity in CKD in T2DM patients may partially be explained by other co-occurring risk factors for kidney damage, such as hypertension, obesity, or dyslipidemia. The analysis of the GWAS data of the Framingham Heart Study cohort was the first to suggest an underlying genetic component of the kidney function in a community-based population [78]. A growing number of GWAS studies in larger and better-defined cohorts of T2DM patients, inclusion of large-scale biobanks, and the aggregation and meta-analysis of diabetes cohorts across the world enabled the researchers to overcome the problem of phenotypic heterogeneity and supported the search for genetic components of diabetes complications, including DKD [79]. Currently, the GWAS catalogue lists 50 datasets for diabetic nephropathy with 199 significant associations. However, with the listed discovery cohorts being mostly of European ancestry and individual polymorphisms having only small effects, it is not surprising that most of the associations were not replicated in validation cohorts of different ethnic backgrounds [80]. For example, rs136161 located in the Apolipoprotein L1 coding gene (APOL1), one of the loci linked to DKD across multiple ethnic groups [81], was recently replicated in a relatively small study cohort from the Genome Database of the Latvian Population [82] but was not replicated in another small cohort from the Emirati population [83].

Identification of the susceptible loci associated with CKD risk may support early identification of T2DM patients at increased risk, while identification of SGLT2 inhibitors targets may support targeted therapeutic interventions through the analysis of a panel of genetic markers. The challenge of translating the risk loci identified by GWAS into clinical practice was solved by the introduction of the polygenic risk score (PRS), which presents the weighted sum of risk alleles, taking into account their effect sizes [84]. With the larger GWAS sample sizes and advanced computational algorithms, the prediction value of the PRS in high-risk groups was predicted to be similar to the prediction value of positive genetic tests for developing monogenic diseases [85]. PRS for multiple phenotypes (for example, diabetes, obesity, albuminuria, and others) associated with the development of diabetic complications can be combined into so-called multi-PRS, which were shown to have a better predictive value compared to individual PRS. Such multi-PRS models supported stratification of T2DM patients according to the risk of developing late complications and identification of patients who may benefit from more intensive treatment [86]. However, PRS are strongly dependent on the population’s genetic background, which may potentially lead to health disparities in ethnic groups that are under-represented in GWAS [87].

The limitations of the GWAS approaches in their ability to capture only common genetic variations with a frequency ≥1% may be overcome by whole-exome or whole-genome sequencing that have the potential of detecting rare genetic variants, among them also variants with potentially larger effect sizes, that may contribute to CKD [88].

The observation of pleiotropic effects of SGLT2i on both heart and kidney failure led to the search for overlapping genetic backgrounds between these two late complications and for the potential common targets of SGLT2 inhibitors. Li et al. used canonical correlation analysis based on multivariate regression analysis on the summary statistics data from two large independent meta-analyses of GWAS. This led to identification of 4624 SNPs and 1745 genes associated with CKD and heart failure (HF). The identified genes were validated against the transcriptome-wide gene expression data for CKD and HF. Next, gene enrichment and KEGG pathway analyses were used to explore the potential functional significance of the identified genes and targets. This led to 169 putative pleiotropic genes significantly associated with at least one of the two conditions. Among these genes, 21 were predicted as potential therapeutic targets of SGLT2 inhibitors in both CKD and HF [89].

### 5.2. Epigenomic Factors Related to CKD and SGLT2 Action

Although genetic factors may contribute to the development, progression, and clinical manifestations of T2DM and its complications, they more likely result from complex interactions between genetic, metabolic, and environmental factors. Furthermore, metabolic and environmental factors may alter gene expression levels by tissue-specific epigenetic modifications, such as changes in CpG methylation, posttranslational histone modifications, and changes in expression levels of non-coding RNAs. Periods of high glucose levels shown to lead to tissue-specific epigenetic modifications in various tissues have recently emerged as additional mediators of microvascular complications [90].

The analysis of DNA methylation in proximal kidney tubule cells in a mouse model revealed selective hypomethylation of promotor regions of genes involved in glucose metabolism, including SGLT2 in non-diabetic mice, as compared to gene-specific changes in methylation levels and also hypermethylation of several other metabolic and transporter genes in diabetic mice. These changes in methylation levels were accompanied by histone modification changes as well as changes in gene expression levels. This altered DNA methylation pattern observed in diabetic mice did not reverse under pioglitazone treatment as it led to persistent alterations in mRNA expression levels [91]. This long-term persistence in epigenetic changes may explain why even transient periods of high glucose levels may elicit long-term changes in gene expression levels that may lead to the development of chronic T2DM complications. Rapid technological advancement has also enabled epigenome-wide association studies (EWAS). A study investigating genome-wide cytosine methylation profiling of tubule epithelial cells obtained from CKD and control kidneys observed that differentially methylated regions overlap with enhancer regions of important renal transcription factors and altered gene expression levels of genes related to kidney fibrosis, thus directly linking epigenetic changes with CKD [92].

A recent meta-analysis of EWAS from five European cohorts identified 76 CpG sites differentially methylated in individuals with T2DM compared with control individuals. Although many of these sites were linked to insulin signaling, lipid homeostasis, and inflammation, the inclusion of the methylation status did not improve the T2DM risk prediction models when compared to the models based on established predictors, such as age, sex, BMI, and HbA_1c_ [93]. The level of DNA methylation was also altered in peripheral blood leukocytes of T2DM patients with late complications. Lower levels of genomic DNA methylation were significantly associated with the risk of diabetic polyneuropathy but were not associated with DKD or other chronic T2DM complications [94]. As shown in a recent EWAS study in an Indian population, specific methylation patterns can be used as biomarkers for stratification of T2DM patients with different clinical outcomes as they enable separating patients with good glycemic control and dyslipidemia from the other patients [95].

Several classes of drugs that may target epigenetic changes are being developed with the aim to reverse disease-related epigenetic changes, such as drugs inhibiting DNA methyltransferases, histone acetyl transferases, histone deacetylases, protein methyltransferases, and histone methyltransferases [96,97]. It has been shown that SGLT2 inhibitors may also act as epigenetic modifiers [98]. As SGLT2 inhibitors treatment leads to increased circulating and tissue levels of β-hydroxybutyrate, this may consequently increase the level of histone H3 β-hydroxybutyrylation [99]. In adipocytes, the adiponectin gene was shown to be regulated by β-hydroxybutyrylation independently of acetylation or methylation status. Due to its anti-inflammatory and anti-atherogenic properties, the pleiotropic protective effect of SGLT2 inhibitors may be partially elicited via their effect on epigenetic regulation [100].

Another epigenetic mechanism that may be affected by SGLT2 inhibitors is microRNA (miRNA) levels. MiRNAs regulate gene expression at the post-transcriptional level by binding to target mRNAs and silencing their translation. Dysregulation of miRNA expression may trigger the molecular processes of diseases. On the other hand, disease-related processes may lead to dysregulation in miRNA expression. As miRNAs are secreted from the cells into bodily fluids, they are a very promising non-invasive biomarker of disease-related processes. It has been shown in DKD that miRNA expression is dysregulated within the kidney tissue as well as in blood plasma, and methods have rapidly evolved for isolation and analysis of soluble miRNA and miRNA contained within the extracellular vesicles. As membrane antigens and intracellular proteins of extracellular vesicles enable identification of their cells of origin, the analysis of miRNAs in the context of extracellular vesicles, such as small exosomes, may provide even more information on disease pathogenesis than soluble miRNA [101].

In DKD, specific miRNAs have been found to be dysregulated both within the kidney tissue [102] as well as in plasma, where specific miRNAs enriched in extracellular vesicles may constitute a non-invasive biomarker of diabetes complications [103,104].

In addition to their potential role as diagnostic and prognostic biomarkers, miRNA may also serve as predictive biomarkers of treatment response. Namely, their specificity and expression levels may also change in response to treatment and are reflected in the circulating miRNA levels. For example, in an open-label study comparing treatment with dapagliflozin or thiazides in 40 T2DM patients, changes occur in expression levels of circulating miRNAs previously described as associated with heart failure, endothelial dysfunction, and kidney damage after 4 weeks of treatment with dapagliflozin, again proving that the protective effects of SGLT2 inhibitors on cardiorenal function may be mediated by epigenetic changes and that specific miRNAs could be used as predictive biomarkers of treatment response [105].

### 5.3. Proteomic and Inflammatory Biomarkers

The nephroprotective effects of SGLT2 inhibitors may also be related to their anti-inflammatory effects mediated by decreased circulating levels of inflammatory factors, such as IL-6, TNF receptor-1, matrix metalloproteinase-7, and fibronectin-1, as well as nephroprotective metabolic effects due to reduced ketone production [106]. The involvement of oxidative stress related to metabolic factors, inflammation, fibrotic factors, and hemodynamic factors in the molecular pathogenesis of DKD was supported by a large number of studies (recently reviewed in 103); however, the molecular mechanisms of how SGLT2 inhibitors target these pathways need to be better elucidated. The studies that assessed inflammatory factors before and after SGLT2 inhibitors treatment are very few as most clinical trials focused on direct glucose-lowering effects, clinical outcomes, and kidney function measures.

Dekkers et al. investigated the effect of dapagliflozin treatment on the levels of markers of glomerular (IgG to IgG4 and IgG to albumin ratio) and tubular damage (urinary KIM-1, NGAL, and LFABP) and inflammation (urinary MCP-1 and IL-6) in urine samples obtained from 33 T2DM patients during a prospective, randomized, double-blind, placebo-controlled cross-over clinical trial of dapagliflozin versus placebo. They have shown that 6 weeks of treatment with dapagliflozin decreases levels of KIM-1 as a marker of proximal tubular damage as well as levels of inflammatory marker IL-6 in the urine and that the reduction in these markers correlated with the lowering of albuminuria [49].

Heerspink et al. used systems medicine approaches supported by omics technologies to explain molecular mechanisms of disease-related processes and SGLT2 inhibitors treatment response and provided direct evidence regarding how canagliflozin affects the biomarkers related with progression of DKD [107]. Based on the literature and the publicly available transcriptomic data from in vitro experiments in human proximal tubular cells, they first constructed bioinformatic models of the molecular mechanism of canagliflozin action of SGLT inhibitors. They overlapped this model with a previously published model of the molecular mechanisms of DKD [108]. The bioinformatic analysis identified 44 proteins that overlapped between both models, and these proteins were selected as candidate biomarkers for monitoring the effects of canagliflozin treatment on DKD. In a validation study, they measured these overlapping proteins in 296 plasma samples of T2DM patients available from the Canagliflozin Treatment and Trial Analysis-Sulfonylurea (CANTATA-SU) trial, a phase 3 clinical trial comparing canagliflozin with glimepiride [49]. The study provided direct evidence that canagliflozin reduced plasma levels of inflammatory factors, such as IL-6, TNF receptor-1 (TNFR1), matrix metalloproteinase-7 (MMP7), and fibronectin-1 (FN1) [109].

Another study measured the levels of three candidate biomarkers of kidney injury in longitudinally collected plasma samples of T2DM patients obtained at baseline and years 1, 3, and 6 after inclusion in the CANVAS trial. They showed that, over time and in F comparison to placebo, canagliflozin decreased the plasma levels of kidney injury molecule 1 (KIM1) and modestly attenuated the rise of TNFR1 and TNFR2, and that the early reductions in TNFR-1 and TNFR-2 were independently associated with a lower risk of DKD progression [110].

The anti-inflammatory effects of SGLT2 inhibitors were also ascribed to their effects on the increased ketogenesis and increased plasma ketone levels [111]. Despite the concerns related to the risk of developing diabetic ketoacidosis, increased ketone levels were actually shown to modulate the activity of NLRP3 inflammasome, leading to reduced secretion of IL-1β in macrophages of T2DM patients with high cardiovascular risk after 30 days of empagliflozin treatment as compared to the levels in macrophages of patients undergoing sulfonylurea treatment [112]. These findings were consistent with data from a diabetic mouse model showing that inhibition of NLRP3 inflammasome suppressed inflammatory response and oxidative stress and improved diabetic nephropathy [113].

All the described studies indicate that these inflammatory molecules, and, in particular, IL-6, are both key mediators and prospective biomarkers of DKD development.

## 6. Conclusions

Although the genetic background of T2DM and its complications is firmly confirmed, evidence suggests that there may also be a genetic component to the response to SGLT2 inhibitors. However, the pleiotropic protective effects of SGLT2 inhibitors that extend beyond their glucose-lowering effects of treatment can also be partially ascribed to epigenetic factors and accumulation of the aberrant metabolites that may activate stress response and inflammation and result in a spectrum of clinical phenotypes. It is generally accepted that interactions between genetic, epigenetic, metabolic, and environmental factors may modify the disease risk, the course of these diseases, and treatment outcomes. A better understanding of these molecular mechanisms and their interactions may lead to the discovery of diagnostic, prognostic, and predictive biomarkers that would enable early recognition of individuals at risk for the development of disease or disease-related complications, unfavorable prognosis, or aberrant treatment response and serve as the basis for the application of Predictive, Preventive, Personalized, and Participatory (P3) medicine.

## Figures and Tables

**Figure 1 pharmaceutics-15-01995-f001:**
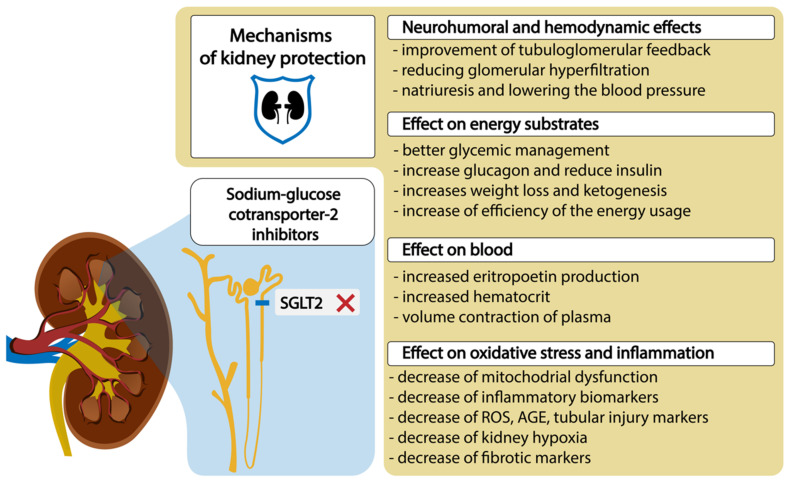
SGLT2 inhibitors and mechanisms of diabetic kidney protection. Abbreviations: ROS, reactive oxygen species; AGE, advanced glycolytic end product.

## Data Availability

Data used for the writing of this review are available from the authors on request.

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
