# Peer review of "SGLT2 Inhibitors in the Treatment of Diabetic Kidney Disease: More than Just Glucose Regulation"

_pharmaceutics, 2023, doi:10.3390/pharmaceutics15071995_

Round 1

Reviewer 1 Report

Dear Editor, I’ve read with great interest the draft called “SGLT2 inhibitors in the treatment of diabetic kidney disease: 2 more than just glucose regulation” by Jasna Klen and Vita Dolžan. However, some issues need to be raised.

-          Line 14: Please insert “mellitus” and throughout the paper.

-          Line 58: A recent original article demonstrated that the increase in the number of risk factors at target correlates with better cardiovascular-free survival in patients with type 2 diabetes and microalbuminuria. I would suggest seeing and briefly discussing Sasso et al. The number of risk factors not at target is associated with cardiovascular risk in a type 2 diabetic population with albuminuria in primary cardiovascular prevention. Post-hoc analysis of the NID-2 trial. Cardiovasc Diabetol. 2022 Nov. doi: 10.1186/s12933-022-01674-7.

-          Line 217: typing error “addition”.

-          Line 223: Typing error (double space).

-          Line 247: typing error “redutions”.

-          Line 288: typing error.

-          Line 311-312: a table legend is needed.

-          Line 371: typing error.

-          I would suggest moving the paragraph “4. Treatment with SGLT2 inhibitors and kidney outcomes”, as last paragraph before “6. Conclusions”.

Reviewer 2 Report

This is a well-written review that provides novel data on the cardiorenal protection afforded by SGLT-2 inhibitors in patients with type 2 diabetes.

Some comments to the authors:

1.       The clinical-trial data should be presented better. The authors describe together evidence from cardiovascular outcome trials, trials conducted in patients with heart failure and trials conducted in patients with heart failure. This mixture generates confusion.

2.       There is a new meta-analysis published in Lancet that provides an overall estimate of the treatment effects of SGLT-2 inhibitors in patients with or without type 2 diabetes. Please include these data.

3.       The beneficial effect of SGLT-2 inhibitors on the risk of acute kidney injury needs to be discussed. Despite the concerns, these agents appear to reduce the incidence of AKI. Which are the mechanisms?

4.       The favorable effect of SGLT-2 inhibitors on serum potassium levels in also important. These agents can mitigate the risk of hyperkalemia to enable the concomitant use of an MRA.

5.       Which is the future in the treatment of DKD? The authors need to mention that finerenone is another effective therapy that has received regulatory approval for cardiorenal protection in patients with type 2 diabetes. There is also the SLOW trial that investigates the potential kidney protective effects of GLP1-RAs in patients with DKD.

Reviewer 3 Report

The present manuscript evaluates the implications of iSGLT-2 in the treatment of chronic kidney disease. Although the subject has been previously addressed, the present review discusses also the implication of other contributing factors such as genetic, epigenetic, transcriptomic and proteomic. By adding new relevant dat regarding this aspects, the quality of the manuscript is enhanced. 

The figures are adequate. The conclusion are in accordance with data presented. 

The keywords could be adapted according to MeSH on demand for easier article indexing. There is a small error in the Abstract that can be rephrased:  "decrease serum creatinine levels doubling". 

Round 2

Reviewer 2 Report

The authors have to include a seperate section with the aim to discuss the risk of AKI with SGLT-2 inhibitor therapy. This is very important.

The authors could also include the following article in this review: Int J Mol Sci. 2023 Feb 1;24(3):2803. doi: 10.3390/ijms24032803.
PMID: 36769113.
